# Deep Rao-Blackwellised Particle Filters for Time Series Forecasting

**Richard Kurle**[*†2] **Syama Sundar Rangapuram**[1] **Emmanuel de Bezenac**[†3]
**Stephan Günnemann**[2]     **Jan Gasthaus**[1]
[1]AWS AI Labs     [2]Technical University of Munich     [3]Sorbonne Université

## Abstract

This work addresses efficient inference and learning in switching Gaussian linear dynamical systems using a Rao-Blackwellised particle filter and a corresponding Monte Carlo objective. To improve the forecasting capabilities, we extend this classical model by conditionally linear state-to-switch dynamics, while leaving the partial tractability of the conditional Gaussian linear part intact. Furthermore, we use an auxiliary variable approach with a decoder-type neural network that allows for more complex non-linear emission models and multivariate observations. We propose a Monte Carlo objective that leverages the conditional linearity by computing the corresponding conditional expectations in closed-form and a suitable proposal distribution that is factorised similarly to the optimal proposal distribution. We evaluate our approach on several popular time series forecasting datasets as well as image streams of simulated physical systems. Our results show improved forecasting performance compared to other deep state-space model approaches.

## 1   Introduction

The Gaussian linear dynamical system (GLS) [4, 38, 31] is one of the most well-studied dynamical models with wide-ranging applications in many domains, including control, navigation, and time-series forecasting. This state-space model (SSM) is described by a (typically Markovian) latent linear process that generates a sequence of observations. The assumption of Gaussian noise and linear state transitions and measurements allows for exact inference of the latent variables—such as filtering and smoothing—and computation of the marginal likelihood for system identification (learning). However, most systems of practical interest are non-linear, requiring more complex models.

Many approximate inference methods have been developed for non-linear dynamical systems: Deterministic methods approximate the filtering and smoothing distributions e.g. by using a Taylor series expansion of the non-linear functions (known as extended Kalman filter (EKF) and second-order EKF) or by directly approximating these marginal distributions by a Gaussian using moment matching techniques [26, 27, 2, 3, 36]. Stochastic methods such as particle filters or smoothers approximate the filtering and smoothing distributions by a set of weighted samples (particles) using sequential Monte Carlo (SMC) [13, 16]. Furthermore, system identification with deep neural networks has been proposed using stochastic variational inference [17, 15] and variational SMC [23, 29, 20].

A common challenge with both types of approximations is that predictions/forecasts over long forecast horizons suffer from accumulated errors resulting from insufficient approximations at every time step. Switching Gaussian linear systems (SGLS) [1, 12]—which use additional latent variables to switch between different linear dynamics—provide a way to alleviate this problem: the conditional linearity can be exploited by approximate inference algorithms to reduce approximation errors. Unfortunately, this comes at the cost of reduced modelling flexibility compared to more general non-linear dynamical systems. Specifically, we identify the following two weaknesses of the SGLS: i) the switch transition

---

[*]Correspondence to `richard.kurle@tum.de`. [†]Work done while at AWS AI Labs.

model is assumed independent of the GLS state and observation; ii) conditionally-linear emissions are insufficient for modelling complex multivariate data streams such as video data, while more suitable emission models (e.g. using convolutional neural networks) exist. The first problem has been addressed in [21] through augmentation with a Polya-gamma-distributed variable and a stick-breaking process. However, this approach uses Gibbs sampling to infer model parameters and thus does not scale well to large datasets. The second problem is addressed in [11] using an auxiliary variable between GLS states and emissions and stochastic variational inference to obtain a tractable objective function for learning. Yet, this model predicts the GLS parameters deterministically using an LSTM with an auto-regressive component, resulting in poor long-term forecasts.

We propose auxiliary variable recurrent switching Gaussian linear systems (ARSGLS), an extension of the SGLS to address both weaknesses by building on ideas from [21] and [11]. ARSGLS improves upon the SGLS by incorporating a conditionally linear state-to-switch dependence, which is crucial for accurate long-term out-of-sample predictions (forecasting), and a decoder-type neural network that allows modelling multivariate time-series data with a non-linear emission/measurement process. As in the SGLS, a crucial feature of ARSGLS is that approximate inference and likelihood estimation can be Rao-Blackwellised, that is, expectations involving the conditionally linear part of the model can be computed in closed-form, and only the expectations wrt. the switch variables need to be approximated. We leverage this feature and propose a Rao-Blackwellized filtering objective function and a suitable proposal distribution for this model class.

We evaluate our model with two different instantiations of the GLS: in the first scenario, the GLS is implemented with a constrained structure that models time-series patterns such as level, trend and seasonality [14, 34]; the second scenario considers a general, unconstrained conditional GLS. We compare our approach to closely related methods such as Deep State Space Models (DeepState) [30] and Kalman Variational Auto-Encoders (KVAE) [11] on 5 popular forecasting datasets and multivariate (image) data streams generated from a physics engine as used in [11]. Our results show that the proposed model achieves improved performance for univariate and multivariate forecasting.

## 2 Background

### 2.1 Gaussian Linear Dynamical Systems

The GLS models sequence data using a first-order Markov linear latent and emission process:

$$\mathbf{x}_t = A\mathbf{x}_{t-1} + B\mathbf{u}_t + \mathbf{w}_t, \qquad \mathbf{w}_t \sim \mathcal{N}(0, R), \tag{1a}$$

$$\mathbf{y}_t = C\mathbf{x}_t + D\mathbf{u}_t + \mathbf{v}_t, \qquad \mathbf{v}_t \sim \mathcal{N}(0, Q), \tag{1b}$$

where the vectors $\mathbf{u}$, $\mathbf{x}$ and $\mathbf{y}$ denote the inputs (also referred to as covariates or controls), latent states, and targets (observations). $A$, and $C$ are the transition and emission matrix, $B$ and $D$ are optional control matrices, and $R$, $Q$ are the state and emission noise covariances. In the following, we will omit the optional inputs $\mathbf{u}$ to ease the presentation, however, we use inputs e.g. for our experiments in Sec. 5.2. The appealing property of the GLS is that inference and likelihood computation is analytically tractable using the well-known Kalman filter algorithm, alternating between a prediction step $p(\mathbf{x}_t \mid \mathbf{y}_{1:t-1}) = \int p(\mathbf{x}_t \mid \mathbf{x}_{t-1})p(\mathbf{x}_{t-1} \mid \mathbf{y}_{1:t-1})d\mathbf{x}_{t-1}$ and update step $p(\mathbf{x}_t \mid \mathbf{y}_{1:t}) \propto p(\mathbf{x}_t \mid \mathbf{y}_{1:t-1})p(\mathbf{y}_t \mid \mathbf{x}_t)$, where $p(\mathbf{x}_t \mid \mathbf{x}_{t-1})$ is the state transition and $p(\mathbf{y}_t \mid \mathbf{x}_t)$ is the emission/measurement process corresponding to Eqs. (1a) and (1b), respectively.

### 2.2 Particle Filters

In non-linear dynamical systems, the filter distribution $p(\mathbf{x}_t \mid \mathbf{y}_{1:t})$ is intractable and needs to be approximated. Particle filters are SMC algorithms that approximate the filter distribution at every time-step $t$ by a set of $P$ weighted particles $\{\mathbf{x}^{(p)}\}^P$, combining importance sampling and re-sampling. Denoting the Dirac measure centred at $\mathbf{x}_t$ by $\delta(\mathbf{x}_t)$, the filter distribution is approximated as

$$p(\mathbf{x}_t \mid \mathbf{y}_{1:t}) \approx \sum_{p=1}^{P} w_t^{(p)} \delta(\mathbf{x}_t^{(p)}). \tag{2}$$

In first-order Markovian dynamical systems, the importance-weights are computed recursively as

$$\tilde{w}_t^{(p)} = w_{t-1}^{(p)} \gamma(\mathbf{x}_t^{(p)}, \mathbf{x}_{t-1}^{(p)}), \quad w_t^{(p)} = \frac{\tilde{w}_t^{(p)}}{\sum_{p=1}^{P} \tilde{w}_t^{(p)}}, \quad \gamma(\mathbf{x}_t^{(p)}; \mathbf{x}_{t-1}^{(p)}) = \frac{p(\mathbf{y}_t \mid \mathbf{x}_t^{(p)})p(\mathbf{x}_t^{(p)} \mid \mathbf{x}_{t-1}^{(p)})}{\pi(\mathbf{x}_t^{(p)} \mid \mathbf{x}_{1:t-1}^{(p)})}, \tag{3}$$

where $\tilde{w}_t^{(p)}$ and $w_t^{(p)}$ denote the unnormalised and normalised importance-weights, respectively, $\gamma(\mathbf{x}_t^{(p)}; \mathbf{x}_{t-1}^{(p)})$ is the incremental importance-weight, and $\pi(\mathbf{x}_t \mid \mathbf{x}_{1:t-1}^{(p)})$ is the proposal distribution. To alleviate weight degeneracy issues, a re-sampling step is performed if the importance-weights satisfy a chosen degeneracy criterion. A common criterion is to re-sample when the effective sample size (ESS) drops below half the number of particles, i.e. $P_{\text{ESS}} = \left(\sum_{p=1}^{P}(w^{(p)})^2\right)^{-1} \leq P/2$.

## 2.3 Switching Gaussian Linear Systems with Rao-Blackwellised Particle Filters

Switching Gaussian linear systems (SGLS), also referred to as conditional GLS or mixture GLS, are a class of non-linear SSMs that use additional (non-linear) latent variables $\mathbf{s}_{1:t}$ that index the parameters (transition, emission, control, and noise covariance matrices) of a GLS, allowing them to "switch" between different (linear) regimes:

$$
\begin{aligned}
\mathbf{x}_t &= A(\mathbf{s}_t)\mathbf{x}_{t-1} + B(\mathbf{s}_t)\mathbf{u}_t + \mathbf{w}_t(\mathbf{s}_t), & \mathbf{w}_t(\mathbf{s}_t) &\sim \mathcal{N}(0, R(\mathbf{s}_t)), \\
\mathbf{y}_t &= C(\mathbf{s}_t)\mathbf{x}_t + D(\mathbf{s}_t)\mathbf{u}_t + \mathbf{v}_t(\mathbf{s}_t), & \mathbf{v}_t(\mathbf{s}_t) &\sim \mathcal{N}(0, Q(\mathbf{s}_t)).
\end{aligned}
\tag{4}
$$

The switch variables $\mathbf{s}_t$ are typically categorical variables, indexing one of $K$ base matrices; however, other choices are possible (see Sec. 3.1). Omitting the inputs $\mathbf{u}$ again, the graphical model factorises as $p(\mathbf{y}_{1:T}, \mathbf{x}_{0:T}, \mathbf{s}_{1:T}) = p(\mathbf{x}_0)\prod_{t=1}^{T} p(\mathbf{y}_t \mid \mathbf{x}_t, \mathbf{s}_t)p(\mathbf{x}_t \mid \mathbf{x}_{t-1}, \mathbf{s}_t)p(\mathbf{s}_t \mid \mathbf{s}_{t-1})$, where $p(\mathbf{s}_1 \mid \mathbf{s}_0) = p(\mathbf{s}_1)$ is the switch prior (conditioned on $\mathbf{u}_1$ if inputs are given). Note that we use an initial state $\mathbf{x}_0$ without corresponding observation for convenience.[2]

A crucial property of this model is that, while the complete model exhibits non-linear dynamics, given a sample of the trajectory $\mathbf{s}_{1:T}^{(p)}$, the rest of the model is (conditionally) linear. This allows for efficient approximate inference and likelihood estimation. The so-called Rao-Blackwellised particle filter (RBPF) leverages the conditional linearity by approximating expectations—that occur in filtering and likelihood computation—wrt. the switch variables $\mathbf{s}_{1:T}$ using SMC, while computing expectations wrt. the state variables $\mathbf{x}_{1:T}$ analytically. To achieve this, the posterior distribution is factorised as

$$
p(\mathbf{x}_{1:t}, \mathbf{s}_{1:t} \mid \mathbf{y}_{1:t}) = p(\mathbf{x}_{1:t} \mid \mathbf{s}_{1:t}, \mathbf{y}_{1:t})p(\mathbf{s}_{1:t} \mid \mathbf{y}_{1:t}).
\tag{5}
$$

The first term of Eq. (5) is tractable given a sample trajectory $\mathbf{s}_{1:t}^{(p)}$ using a Kalman smoother. However, for forecasting and loss computation (Sec. 3.3.2) we only require the state from the last step $p(\mathbf{x}_t \mid \mathbf{s}_{1:t}, \mathbf{y}_{1:t})$ for which even the Kalman filter suffices (cf. supplementary material for details). The second term of Eq. (5) can also be computed recursively using Bayes rule:

$$
p(\mathbf{s}_{1:t} \mid \mathbf{y}_{1:t}) \propto p(\mathbf{y}_t \mid \mathbf{s}_{1:t}, \mathbf{y}_{1:t-1})\, p(\mathbf{s}_t \mid \mathbf{s}_{t-1}, \cancel{\mathbf{s}_{1:t-2}}, \cancel{\mathbf{y}_{1:t-1}})\, p(\mathbf{s}_{1:t-1} \mid \mathbf{y}_{1:t-1}).
\tag{6}
$$

Here and in the following we explicitly show the terms that cancel due to the Markov-property. The predictive distribution

$$
p(\mathbf{y}_t \mid \mathbf{s}_{1:t}, \mathbf{y}_{1:t-1}) = \int p(\mathbf{y}_t \mid \mathbf{x}_t, \mathbf{s}_t, \cancel{\mathbf{s}_{1:t-1}}, \cancel{\mathbf{y}_{1:t-1}})p(\mathbf{x}_t \mid \mathbf{s}_{1:t}, \mathbf{y}_{1:t-1})d\mathbf{x}_t
$$

is a by-product of the first term in Eq. (5); it can be computed in closed-form using the Kalman filter. Since all terms in $p(\mathbf{s}_{1:t} \mid \mathbf{y}_{1:t})$ can be computed in closed-form, this distribution can be approximated by a set of particles using SMC (cf. Eq. (2)). The corresponding incremental importance-weights are

$$
\gamma(\mathbf{s}_t^{(p)}; \mathbf{s}_{1:t-1}^{(p)}) = \frac{p(\mathbf{y}_t \mid \mathbf{s}_{1:t}^{(p)}, \mathbf{y}_{1:t-1})p(\mathbf{s}_t^{(p)} \mid \mathbf{s}_{t-1}^{(p)})}{\pi(\mathbf{s}_t^{(p)} \mid \mathbf{s}_{1:t-1}^{(p)})}.
\tag{7}
$$

The resulting joint filter distribution for both the state and switch variables is a mixture of Gaussians:

$$
p(\mathbf{x}_t, \mathbf{s}_t \mid \mathbf{y}_{1:t}) \approx \sum_{p=1}^{P} w_t^{(p)}\delta(\mathbf{s}_t^{(p)})\, \mathcal{N}\big(\mathbf{x}_t \mid m_t(\mathbf{s}_{1:t}^{(p)}), V_t(\mathbf{s}_{1:t}^{(p)})\big),
\tag{8}
$$

where the history $\mathbf{s}_{1:t-1}^{(p)}$ is dropped. To be precise, the mean $m_t(\mathbf{s}_{1:t}^{(p)})$ and variance $V_t(\mathbf{s}_{1:t}^{(p)})$ of the filtered state $\mathbf{x}_t$ depend on the whole trajectory $\mathbf{s}_{1:t}^{(p)}$, although from an algorithmic perspective, only the current switch $\mathbf{s}_t^{(p)}$ is required to compute the incremental importance-weights in Eq.(7).

# 3 Auxiliary Variable Recurrent Switching Gaussian Linear Systems

The SGLS provides an attractive tradeoff between model complexity and efficient inference. However, this model is limited in the following respects: i) switch transitions are assumed to be independent of the state variable; ii) multivariate observations with a complex non-linear dependence—such as streams of image data—are not well-described by a conditionally linear emission model. We propose the auxiliary variable recurrent SGLS (ARSGLS) to address both of these issues in Secs. 3.1 and 3.2. Furthermore, we leverage the conditional linearity using a RBPF (Sec. 3.3) with a suitable proposal distribution that is structurally simliary to the optimal (minimum variance) proposal distribution. The resulting graphical model and corresponding proposal distribution is visualised in Fig. 1 and the algorithm for filtering and loss computation is presented in Alg. 1 in the supplementary material.

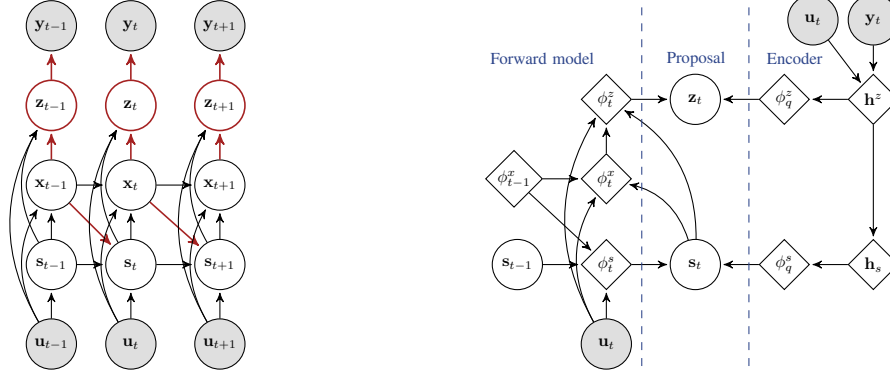

Figure 1: Left: Forward graphical model of ARSGLS, incl. inputs (controls) $\mathbf{u}_t$. Shaded/unshaded nodes indicate observed/hidden variables. New components (auxiliary variable, state-to-switch recurrence) are marked in red. Right: Inference proposal distribution for time step $t$. The left part is identical (shared) to the forward model, the right part is the approximated likelihood function given by a ladder-type encoder, and the proposal distribution results from taking the product of the two respective Gaussian functions (cf. Sec. 3.3.3, Eq. (14)). To visualise the marginalisation of $\mathbf{x}_{t-1}$ and $\mathbf{x}_t$ as well as the product of Gaussian functions of the variables $\mathbf{s}_t$ and $\mathbf{z}_t$, we explicitly denote the distribution parameters (location, covariance) of the respective Gaussians by $\phi$.

## 3.1 Gaussian Recurrent Switch Transition

Although the classical SGLS is capable of generating sequences with non-linear dynamics (due to non-linear switches), it makes a very limiting assumption: the switch variables $\mathbf{s}_{1:T}$ are independent of the state variables $\mathbf{x}_{1:T}$, resulting in open loop switch dynamics. However, most non-linear systems can be approximated by a conditionally linear system only through linearisation *at the current state*. Thus, a feedback (closed loop) coupling with the states $\mathbf{x}_{t-1}$ would be necessary (cf. Fig. 2 for an example with a simple non-linear system). Unfortunately, using a more powerful switch transition model $p(\mathbf{s}_t \mid \mathbf{s}_{t-1}, \mathbf{x}_{t-1})$ complicates inference [5, 21]: In order to compute $p(\mathbf{s}_{1:t} \mid \mathbf{y}_{1:t}) = \int p(\mathbf{s}_{1:t}, \mathbf{x}_t \mid \mathbf{y}_{1:t}) d\mathbf{x}_t$, the previous state variable must be integrated out in closed-form (since we want to avoid sampling the states). While this problem has been addressed in [21], we take a different route that admits a re-parametrised proposal distribution, suitable for optimisation with stochastic gradient descent. It has been shown that Gaussian-distributed switch variables [6] can perform as well or better than continuous relaxations of categorical variables [24] in this setting. We propose to use Gaussian switch variables with a learnable Gaussian prior $p(\mathbf{s}_1)$ and conditionally linear state-to-switch transformations that allow for closed-form marginalisation of the state variable:

$$\mathbf{s}_t^{(p)} = F(\mathbf{s}_{t-1}^{(p)})\mathbf{x}_{t-1} + f(\mathbf{s}_{t-1}^{(p)}) + \epsilon_t(\mathbf{s}_{t-1}^{(p)}), \qquad \epsilon_t(\mathbf{s}_{t-1}^{(p)}) \sim \mathcal{N}\big(0, S(\mathbf{s}_{t-1}^{(p)})\big), \qquad (9)$$

where $f$ is a non-linear function (neural network), and $F(\mathbf{s}_{t-1})$ and $S(\mathbf{s}_{t-1})$ are the transition and covariance matrix predicted by the previous switch variable. Marginalising the states results in

$$p(\mathbf{s}_t|\mathbf{s}_{1:t-1}^{(p)}) = \mathcal{N}\big(\mathbf{s}_t; F(\mathbf{s}_{t-1}^{(p)})m_{t-1}(\mathbf{s}_{1:t-1}^{(p)}) + f(\mathbf{s}_{t-1}^{(p)}), \ F(\mathbf{s}_{t-1}^{(p)})V_{t-1}(\mathbf{s}_{1:t-1}^{(p)})F^T(\mathbf{s}_{t-1}^{(p)}) + S(\mathbf{s}_{t-1}^{(p)})\big). \quad (10)$$

The SGLS parameters $\{A, B, C, D, R, Q, F, S\}$ can be predicted from the Gaussian $\mathbf{s}_t^{(p)}$ and $\mathbf{s}_{t-1}^{(p)}$ in different ways, e.g. i) as the direct outputs a neural network; or ii) as a weighted average of a set of base matrices, where the weights are predicted by a neural network. We use the latter approach, which has been shown to be effective by previous work [6]. Deterministic weighted averages are also used in [37, 15, 11], while the "direct" approach of predicting the parameters is used e.g. by [30].

## 3.2 Non-linear Multivariate Emission Model

Time-series data is often non-Gaussian or multivariate with a complex dependence on hidden variables (e.g. images, point clouds, discrete data). To model such data, we augment the recurrent SGLS with a Gaussian latent variable $\mathbf{z}_{1:T}$ that decouples the observations from the SGLS through the conditional independence $p(\mathbf{x}_{1:t} \mid \mathbf{s}_{1:t}, \mathbf{z}_{1:t}, \mathbf{y}_{1:t}) = p(\mathbf{x}_{1:t} \mid \mathbf{s}_{1:t}, \mathbf{z}_{1:t})$. The resulting joint distribution is given as

$$p(\mathbf{y}_{1:T}, \mathbf{z}_{1:T}, \mathbf{x}_{0:T}, \mathbf{s}_{1:T}) = p(\mathbf{x}_0) \prod_{t=1}^{T} p(\mathbf{y}_t \mid \mathbf{z}_t) p(\mathbf{z}_t \mid \mathbf{x}_t, \mathbf{s}_t) p(\mathbf{x}_t \mid \mathbf{x}_{t-1}, \mathbf{s}_t) p(\mathbf{s}_t \mid \mathbf{s}_{t-1}, \mathbf{x}_{t-1}).$$

This *auxiliary variable* $\mathbf{z}_t$ enables the use of arbitrary conditional distributions $p(\mathbf{y}_t \mid \mathbf{z}_t)$ for which the distribution parameters are predicted by a non-linear emission model such as a neural network. Samples of the auxiliary variables can be interpreted as pseudo-observations in the SGLS. This emission model and auxiliary is similar to the additional latent variable in the KVAE [11]; cf. Sec. 4 for further details on the similarities and differences.

## 3.3 Inference and Parameter Estimation

We extend the RBPF from Sec. 2.3 to infer the latent variables $\{\mathbf{s}, \mathbf{x}, \mathbf{z}\}$; and we use maximum likelihood estimation to learn the model parameters $\theta$ (shared between multiple time-series). These include i) the base matrices of the (recurrent) SGLS, ii) the parameters of the state prior, iii) the switch prior and transition (neural network), and iv) the auxiliary variable decoder (neural network).

### 3.3.1 Rao-Blackwellised Particle Filtering

Filtering can be performed analogous to the standard Rao-Blackwellised particle filter (cf. Sec. 2.3). To this end, we factorise the posterior distribution similarly as was shown for the SGLS:

$$p(\mathbf{z}_{1:t}, \mathbf{x}_{1:t}, \mathbf{s}_{1:t} \mid \mathbf{y}_{1:t}) = p(\mathbf{x}_{1:t} \mid \mathbf{z}_{1:t}, \mathbf{s}_{1:t}, \mathbf{y}_{1:t}) \, p(\mathbf{z}_{1:t}, \mathbf{s}_{1:t} \mid \mathbf{y}_{1:t}). \tag{11}$$

The first term in Eq. (11) can again be computed in closed-form while the second density can be approximated using SMC. The incremental importance-weights for this model are given as

$$\gamma(\mathbf{s}_t^{(p)}, \mathbf{z}_t^{(p)}; \mathbf{s}_{1:t-1}^{(p)}, \mathbf{z}_{1:t-1}^{(p)}) = \frac{p(\mathbf{y}_t \mid \mathbf{z}_t^{(p)}) \, p(\mathbf{z}_t^{(p)} \mid \mathbf{z}_{1:t-1}^{(p)}, \mathbf{s}_{1:t}^{(p)}) \, p(\mathbf{s}_t^{(p)} \mid \mathbf{z}_{1:t-1}^{(p)}, \mathbf{s}_{1:t-1}^{(p)})}{\pi(\mathbf{z}_t^{(p)}, \mathbf{s}_t^{(p)} \mid \mathbf{s}_{1:t-1}^{(p)}, \mathbf{z}_{1:t-1}^{(p)})}. \tag{12}$$

The numerator $p(\mathbf{y}_t, \mathbf{z}_t, \mathbf{s}_t \mid \mathbf{y}_{1:t-1}, \mathbf{z}_{1:t-1}, \mathbf{s}_{1:t-1})$ reveals that the switch transition (last term) with state-to-switch dynamics is no longer Markovian. This is because we marginalise the filtered state $p(\mathbf{s}_t \mid \mathbf{s}_{1:t-1}, \mathbf{z}_{1:t-1}) = \int p(\mathbf{s}_t \mid \mathbf{s}_{t-1}, \mathbf{x}_{t-1}) p(\mathbf{x}_{t-1} \mid \mathbf{z}_{1:t-1}, \mathbf{s}_{1:t-1}) d\mathbf{x}_{t-1}$ that depends on all previous switches and pseudo-observations similarly to the generative case in Eq. (10). As for the SGLS in Sec. 2.3, the conditional distribution of the auxiliary variable (second term) is a by-product of the Kalman filter prediction and update step that is required for computing the first term in Eq. (11):

$$p(\mathbf{z}_t \mid \mathbf{z}_{1:t-1}^{(p)}, \mathbf{s}_{1:t}^{(p)}) = \int p(\mathbf{z}_t \mid \mathbf{x}_t) p(\mathbf{x}_t \mid \mathbf{z}_{1:t-1}^{(p)}, \mathbf{s}_{1:t}^{(p)}) d\mathbf{x}_t,$$

$$p(\mathbf{x}_t \mid \mathbf{z}_{1:t-1}^{(p)}, \mathbf{s}_{1:t}^{(p)}) = \int p(\mathbf{x}_t \mid \mathbf{x}_{t-1}, \mathbf{s}_t^{(p)}) p(\mathbf{x}_{t-1} \mid \mathbf{z}_{1:t-1}^{(p)}, \mathbf{s}_{1:t-1}^{(p)}) d\mathbf{x}_{t-1}.$$

### 3.3.2 Parameter Estimation

For learning the model parameters shared between different time-series we use maximum likelihood estimation and stochastic gradient descent (SGD). In the SMC setting, an unbiased estimator of the marginal likelihood $\hat{p}(\mathbf{y}_{1:T}; \theta) = \prod_{t=1}^{T} \sum_{p=1}^{P} \tilde{w}_t^{(p)}(\theta)$ can be obtained from the unnormalised importance-weights [9] (cf. supplementary material for more details). Consequently, $\mathbb{E}\left[ \log \hat{p}(\mathbf{y}_{1:T}; \theta) \right] \leq \log p(\mathbf{y}_{1:T}; \theta)$ due to Jensen's inequality. Based on this, [29, 23, 20] proposed a tractable lower bound to the log-marginal likelihood that can be optimised with SGD:

$$\mathcal{L}(\mathbf{y}_{1:T}; \theta) = \sum_{t=1}^{T} \log \hat{p}(\mathbf{y}_t \mid \mathbf{y}_{1:t-1}; \theta) = \sum_{t=1}^{T} \log \sum_{p=1}^{P} \tilde{w}_t^{(p)}(\theta) \leq \log p(\mathbf{y}_{1:T}; \theta). \tag{13}$$

We propose to use the same objective function, however, leveraging the conditional linearity of our model, using the incremental importance-weights of Eq. (12) for computing the importance-weights (cf. Eq. (3)).

### 3.3.3 Ladder Proposal Distribution

Choosing a good proposal distribution is essential to obtain reliable estimates with low variance. The optimal (minimum variance) proposal distribution is proportional to the joint distribution $p(\mathbf{y}_t, \mathbf{z}_t, \mathbf{s}_t \mid \mathbf{s}_{1:t-1}, \mathbf{z}_{1:t-1}, \mathbf{y}_{1:t-1})$, i.e. the numerator of the incremental importance-weights. We therefore propose a proposal distribution with similar structure, while reusing the known densities:

$$
\begin{aligned}
\pi(\mathbf{z}_t, \mathbf{s}_t \mid \mathbf{z}_{1:t-1}, \mathbf{s}_{1:t-1}, \mathbf{y}_t) &\propto q(\mathbf{z}_t, \mathbf{s}_t \mid \mathbf{y}_t) p(\mathbf{z}_t, \mathbf{s}_t \mid \mathbf{z}_{1:t-1}, \mathbf{s}_{1:t-1}) \\
&= q(\mathbf{z}_t, \mathbf{s}_t \mid \mathbf{y}_t) p(\mathbf{z}_t \mid \mathbf{z}_{1:t-1}, \mathbf{s}_{1:t}) p(\mathbf{s}_t \mid \mathbf{z}_{1:t-1}, \mathbf{s}_{1:t-1})
\end{aligned}
\tag{14}
$$

The last two terms are the transition of the switches and the resulting auxiliary variable in the generative model; the first term is a Gaussian approximation of the likelihood, predicted by an encoder neural network.[3] Both pairs of Gaussians are combined as a product of experts, resulting in a Gaussian proposal distribution. The switch transition is readily available from the previous step $t-1$, while the predictive distribution for the auxiliary variable requires sampling $\mathbf{s}_t$ first. We therefore propose to structure the encoder with dependencies in the same direction as the generative model, i.e. as presented in Eq. (14). The resulting encoder resembles the encoder in the Ladder VAE [35]. Thus, combining it with the densities from the forward model, we obtain a proposal distribution $\pi(\mathbf{z}_t, \mathbf{s}_t \mid \mathbf{z}_{1:t-1}, \mathbf{s}_{1:t-1}, \mathbf{y}_t) = \pi(\mathbf{s}_t \mid \mathbf{z}_{1:t-1}, \mathbf{s}_{1:t-1}, \mathbf{y}_t) \pi(\mathbf{z}_t \mid \mathbf{z}_{1:t-1}, \mathbf{s}_{1:t}, \mathbf{y}_t)$ that is factorised such that samples can be drawn in the same direction as the generative process, while reusing the predictive quantities of the forward (generative) model. The proposal distribution is optimal if the Gaussian encoder distribution is proportional to the likelihood.

We use the effective sample-size criterion mentioned in Sec. 2.2 with *systematic re-sampling* [8]. Following previous work, we omit the score function estimator term for re-sampled variables, resulting in biased but lower variance gradient estimates. We refer to [23, 29, 20] for a discussion.

## 4 Related work

Extensions of the classical SGLS with a state-to-switch dependence have been proposed in previous work [5, 21, 6]: In [5], the required marginalisation of previous Gaussian states is approximated numerically through sampling. [21] uses logistic stick-breaking for the discrete switches and augments the model with a Polya-Gamma distributed auxiliary variable, rendering the states conjugate to the switch variables such that marginalising the Polya-Gamma variables leaves the original model intact. Inference in this model is accomplished through Gibbs sampling, whereas our work uses a RBPF and SGD for state and parameter inference. Different from these two approaches, [6] uses a Gaussian switch variable. In contrast to our work, inference is performed using stochastic variational inference without exploiting the conditional tractability of the SGLS, instead sampling both states and switches.

Many efficient approximate inference methods have been proposed for the classical SGLS, including approaches using variational inference [12], expectation propagation [39], and a RBPF [10]. Our ARSGLS uses an extension of RBPF that includes the additional auxiliary variable (cf. Sec. 3.2) in the SMC approximation. Learning through Monte Carlo objectives [28] for particle filters has been proposed in [23, 29, 20]. Our objective function is a special case that exploits the conditionally linear structure of the ARSGLS through Rao-Blackwellisation to reduce variance in the estimates.

Parameter learning with a conditional GLS and closed-form inference has been proposed previously in DeepState [30] and the KVAE [11]. Our model differs substantially in the i) graphical model, ii) latent variable inference and iii) parameter learning:
i) DeepState and KVAE can be interpreted as a *deterministic* SGLS, where the GLS parameters are predicted by an RNN, whereas our proposed model uses *probabilistic* switch transitions that receive feedback from the previous state. Furthermore, the GLS parameters in DeepState have a fixed structure that model time-series patterns such as level, trend and seasonality; transition and emission matrices are fixed and the two (diagonal) noise covariance matrices are predicted by the RNN directly. The KVAE uses unrestricted GLS parameters, which are predicted as a weighted average of a set of base matrices, where the weights are given by the RNN. Our proposed model uses a similar weighted

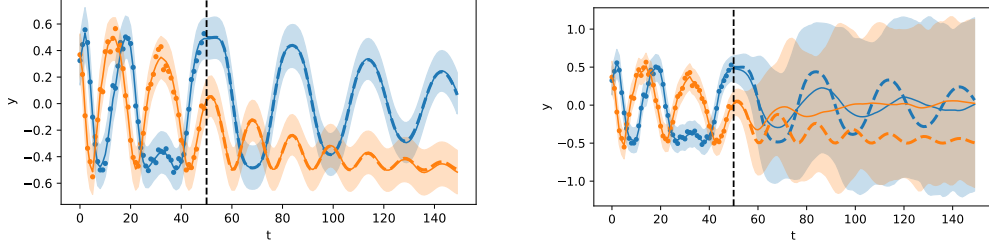

Figure 2: Filtered and forecasted emissions of a recurrent (left) and non-recurrent (right) SGLS for synthetic data of a swinging pendulum. Observations are noisy x-/y-position (blue/orange). The plots show the mean (line) and 3 std. deviations (shaded); filter and forecast range are separated by a vertical line, 50 (noisy) observations (dots) are used for filtering, the forecast ground-truth is shown as a dashed line. The SGLS relies on the filter update, whereas the RSGLS successfully uses state information to select the correct base matrices for prediction.

average of base matrices, but the weights are functions of a *probabilistic* Gaussian switch variable. A further difference to both models is the dependence of the RNN/switch transition on observations and controls: The RNN in DeepState is conditioned on inputs (controls), whereas the RNN in the KVAE depends (auto-regressively) on *samples* of the previous (pseudo-) observation of the GLS. In contrast, the switch transition of the ARSGLS is conditioned on the previous GLS state variable, thus using information from all previous inputs and observations. Finally, the KVAE uses the same type of auxiliary variable and non-linear emission model as our ARSGLS. However, the encoder is conditioned only on the current observations; in contrast, the structure of our proposal distribution (corresponding to the encoder in the KVAE) is chosen to resemble the optimal (minimum variance) proposal distribution, thus including the current observation and inputs as well as the predictive distribution of the previous state.

ii) Given the *deterministically* predicted GLS parameters, DeepState/KVAE uses the Kalman filter/smoother for inference. In contrast, we use a Rao-Blackwellised particle filter to infer non-linear switch variables through SMC and the conditionally linear states through the Kalman filter.

iii) DeepState uses maximum likelihood for parameter learning where the log-likelihood is estimated using the filter formulation (normalisation constants), whereas the KVAE is learnt using a variational EM objective. Our proposed model is learnt through a (Rao-Blackwellised) particle filter-based Monte Carlo objective (cf. prev. paragraph).

## 5 Experiments

### 5.1 Pendulum

We start with a qualitative assessment of the state-to-switch dependence from Sec. 3.1. We refer to this model (without the auxiliary variable of Sec. 3.2) as RSGLS. We consider noisy observations of the xy-positions of a dampened pendulum. Initial states (angle, angular velocity) are drawn randomly from a Gaussian centred at $180 \deg$ (top position) and zero velocity. Both models are trained on 5000 sequences with 50 time-steps. For evaluation, we filter for 50 time-steps and forecast the next 100 time-steps. We visualise the resulting filter and forecast distribution together with noisy observations ($t \leq 50$) and ground-truth ($t > 50$) in Fig. 2. As can be seen, the SGLS can not learn linearised dynamics without information from the state. On the other hand, the RSGLS generates very accurate forecasts with reasonable predictive uncertainty, showing the necessity of state-to-switch dynamics.

### 5.2 Time series forecasting

We evaluate our approach in the context of time-series forecasting on 5 popular public datasets (`electricity`, `traffic`, `solar`, `exchange`, `wiki`) used in [32]. The data is recorded with hourly or daily frequency and exhibits seasonal patterns with different frequency (e.g. daily and weekly).

We experimented with two instantiations of our proposed model: Similar to previous work [30], we implement the GLS as an innovation state space model (ISSM) with a constrained structure that models temporal patterns such as level, trend and seasonality (cf. supplementary material for details). In this model, labelled as RSGLS-ISSM, the dimension of the state $\mathbf{x}_t$ and the transition and emission

|  | CRPS rolling | | | | |
| --- | --- | --- | --- | --- | --- |
|  | exchange | solar | electricity | traffic | wiki |
| DeepAR | 0.009±0.001 | **0.357±0.002** | **0.057±0.003** | **0.120±0.003** | 0.281±0.008 |
| DeepState | 0.010±0.001 | 0.379±0.002 | 0.071±0.000 | 0.131±0.002 | 0.296±0.007 |
| KVAE-MC | 0.010±0.000 | 0.377±0.005 | 0.319±0.010 | 0.233±0.014 | 0.276±0.028 |
| KVAE-RB | 0.009±0.000 | 0.384±0.005 | 0.296±0.024 | 0.179±0.001 | 0.245±0.004 |
| RSGLS-ISSM (ours) | **0.007±0.000** | **0.355±0.004** | 0.070±0.001 | 0.148±0.003 | 0.248±0.006 |
| ARSGLS (ours) | 0.009±0.000 | 0.369±0.008 | 0.138±0.003 | 0.136±0.005 | **0.217±0.010** |
|  | CRPS long-term | | | | |
| DeepAR | 0.019±0.002 | 0.440±0.004 | **0.062±0.004** | 0.138±0.001 | 0.855±0.552 |
| DeepState | 0.017±0.002 | 0.379±0.002 | 0.088±0.007 | **0.131±0.005** | 0.338±0.017 |
| KVAE-MC | 0.020±0.001 | 0.389±0.005 | 0.318±0.011 | 0.261±0.016 | 0.341±0.032 |
| KVAE-RB | 0.018±0.001 | 0.393±0.006 | 0.305±0.022 | 0.221±0.002 | 0.317±0.013 |
| RSGLS-ISSM (ours) | **0.014±0.001** | **0.358±0.001** | 0.091±0.004 | 0.206±0.002 | 0.345±0.010 |
| ARSGLS (ours) | 0.022±0.001 | 0.371±0.007 | 0.154±0.005 | 0.175±0.008 | **0.283±0.006** |

Table 1: CRPS metrics (lower is better). Mean and std. deviation are computed over 4 independent runs for our method and 3 runs for the competing methods. The method achieving the best result is highlighted in **bold**.

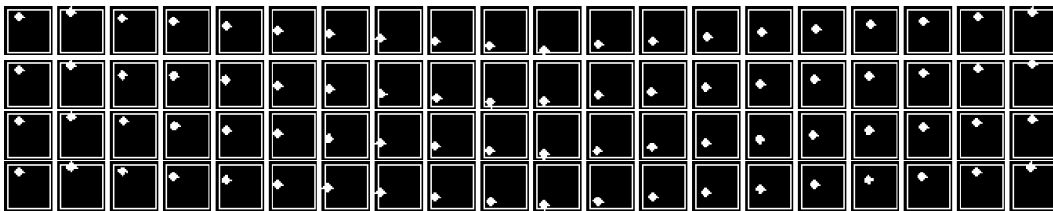

Figure 3: 20 time-steps of groundtruth (1st row) and 3 random trajectories from forecast distribution (remaining rows) for `ARSGLS` on box dataset. Forecasts were generated after 20 filtering time-steps (not shown).

matrices $A$ and $C$ are pre-defined and not learned; for this model, we do not use the non-linear emission model from Sec. 3.2. The second model, labelled `ARSGLS`, uses a general, unrestricted GLS; here we use the complete model, including a non-linear emission model. We compare to two closely related methods, i.e. `DeepState` [30] and `KVAE` [11], and a strong baseline that uses an autoregressive RNN (`DeepAR`) [33]. `DeepState` is implemented with the same ISSM structure as `RSGLS-ISSM`, whereas `KVAE` is implemented with the same unconstrained GLS and measurement model as `ARSGLS`. Furthermore, we note that the objective function proposed in [11] uses *samples* from the smoothing distribution. However, in line with this work, the corresponding expectation can be computed in closed-form; the resulting objective function has lower variance. We implemented both objective functions, denoting them as `KVAE-MC` (Monte Carlo) and `KVAE-RB` (Rao-Blackwellised), respectively.

We use data prior to a fixed forecast date for training and test the forecasts on the remaining data. The probabilistic forecasts are conditioned on the training range and computed with 100 samples for each method. We evaluate in a rolling fashion and with a single long-term forecast. In case of daily/hourly data, the rolling evaluation uses 5/7 windows each with a forecast covering 30 days/24 hours; long-term evaluation uses all 5/7 windows (i.e. 150 days/168 hours) without updating the model with the new data. We score forecasts using the *continuous ranked probability score* (CRPS) [25].

$$\mathrm{CRPS}(F^{-1}, y) = \int_0^1 2\Lambda_\alpha(F^{-1}(\alpha), y)d\alpha, \qquad (15)$$

where $\Lambda_\alpha(q, y) = (\alpha - I_{[y<q]})(y - q)$ is the quantile loss at the quantile level $\alpha \in [0, 1]$, $q$ is the predicted $\alpha$-th quantile level and $y$ is the observation. The results are summarised in Tab. 1. Both of our model variants compare favorably or similarly to their respective closely related competitive model (i.e. `RSGLS-ISSM` and `DeepState`; `ARSGLS` and `KVAE`). Only the auto-regressive baseline `DeepAR` remains challenging on 2/5 datasets.

## 5.3 Simulated physical environments

We evaluate our model for unsupervised forecasting of high-dimensional multivariate data streams (video), emitted from simulated physical environments. To this end, we consider the 4 synthetic

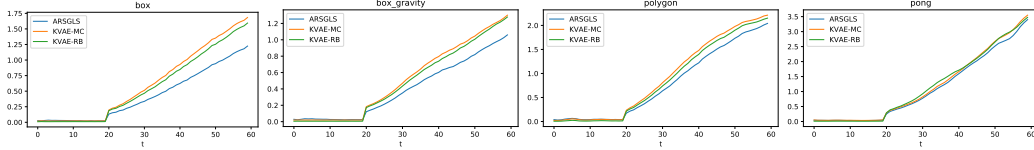

Figure 4: Wasserstein distance in filter/smoothing (`ARSGLS/KVAE`) range ($t < 20$) and forecast range ($t \geq 20$). Results are averaged over 1000 test samples and 3 independent runs.

video datasets (`Box`, `Box-Gravity`, `Polygon`, `Pong`) used in [11], consisting of streams of $32 \times 32$ binary images of an object moving in an environment and colliding with walls. We compare our model to the `KVAE-MC` (proposed in [11]) and our Rao-Blackwellised version `KVAE-RB` cf. 5.2). We use the same model architecture as in [11], except that each model has 10-dimensional states and 10 base matrices (cf. App. 7.7.3 for more details). All models are trained on 5000 time-series of 20 time-steps. We visualise forecasted trajectories in Fig. 3.

Next, we test the learned hidden dynamics quantitatively, comparing the forecasts to the true data. This is challenging for image data: standard metrics such as predictive log-likelihood, (pixel-wise) accuracy or squared error do not differentiate between small and large dis-placements of object in the environment (e.g. if all pixels corresponding to the object are falsely predicted). Interpreting these binary images as a 2D distribution over the xy-positions of the objects in the scene, we propose to score the models using the Wasserstein metric with the Euclidean norm as distance function:

$$\mathrm{D}(\mathbf{y}_\tau, \hat{\mathbf{y}}_\tau) = \int \mathrm{W}(\mathbf{y}_\tau, \hat{\mathbf{y}}_\tau) p(\hat{\mathbf{y}}_\tau \mid \mathbf{y}_{1:t}) d\hat{\mathbf{y}}_\tau, \tag{16}$$

where $p(\hat{\mathbf{y}}_\tau \mid \mathbf{y}_{1:t})$ is the forecast distribution (time-steps $\tau > t$) or predictive distribution from the filter ($t = \tau$). The natural interpretation of this metric is the minimum average distance (in pixel coordinates) needed to move each pixel from the forecast to its true position in the xy-plane. We approximate $p(\hat{\mathbf{y}}_\tau \mid \mathbf{y}_{1:t}) = \int p(\hat{\mathbf{y}}_\tau \mid \mathbf{z}_t, \mathbf{x}_t, \mathbf{s}_t) p(\mathbf{z}_t, \mathbf{x}_t, \mathbf{s}_t \mid \mathbf{y}_{1:t}) d\mathbf{z}_t \mathbf{x}_t \mathbf{s}_t$ (no $\mathbf{s}_t$ in KVAE) using 32 samples and Monte Carlo integration. We evaluate the metric from Eq. (16)—averaged over 3 independent runs and 1000 test data points—for 20 time-steps of filtering (in case of `ARSGLS`) and smoothing (in case of `KVAE`) and 40 time-steps forecasting. Results (Fig. 4) show that `ARSGLS` produces more accurate forecasts.

## 6    Conclusion

In this work, we proposed an extension of the classical SGLS that addresses two weaknesses, while leaving the conditional tractability of this model intact. We improve the forecasting capabilities by conditionally linear state-to-switch recurrence with Gaussian switch variables, keeping the conditional tractability of the GLS intact. Furthermore, we augment the emission model by an auxiliary variable that allows for modelling multi-variate and non-Gaussian observations with complex non-linear transformations such as convolutional neural networks. We leverage the conditional linear structure of this model using Rao-Blackwellised particle filtering and we propose a corresponding Monte Carlo objective for parameter estimation. Experiments on popular time series forecasting datasets and simulated video data from physical environments demonstrate improvements compared to other deep state-space model approaches.

This work offers several interesting research directions: the proposed approach can be generalised to hierarchical models, interleaving linear and non-linear variables; observed data and the corresponding ladder-encoder can be extended to handle multimodal and missing data similar to [19]; offline inference can be done through Rao-Blackwellised particle smoothing [22] and used for learning with a corresponding variational EM objective function; finally, the model parameters may be inferred by means of a variational Bayesian approximation [7] with further extensions to online learning scenarios under data drift [18].

## Broader impact

This paper stems from the author's work on time series forecasting and anomaly detection in industrial settings. The proposed methods are applicable to forecasting from univariate and multivariate data streams more generally. Business applications include supply chain monitoring and sales prediction. Furthermore, accurate forecasts allows better resource management, such as waste reduction and optimisation of energy consumption.

## Funding disclosure

This work was funded by Amazon Research.

## Footnotes

[2]We do not include an initial, unconditional switch variable $\mathbf{s}_0$, as the optimal proposal distribution would be proportional to $p(\mathbf{s}_0, \mathbf{x}_0)$; thus not using observations in the initial proposal distribution would lead to inefficient proposals for the following time steps.

[3]Note that we omitted inputs $\mathbf{u}_t$; however we use both $\mathbf{u}_t$ and $\mathbf{y}_t$ in the encoder. Furthermore, we made the simplifying assumption that $q$ only depends on the latest observation $\mathbf{y}_t$. One way to go beyond that would be to include $\mathbf{y}_{1:t-1}$ through sufficient statistics, such as the natural parameters, of the state variable $\mathbf{x}_{t-1}$. We decided to use only the latest observation $\mathbf{y}_t$ in our model, since information from $\mathbf{x}_{t-1}$ and thus $\mathbf{y}_{1:t-1}$ is incorporated into the proposal distribution through the state-to-switch recurrence.

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
