[Supplementary Material]

# 7 Supplementary Material

## 7.1 Conditional Kalman Filter equations

As mentioned in Sec. 2.3, the first term in Eq. (5) can be computed using the standard Kalman smoother. In this work, we are interested mainly in forecasting and parameter estimation (Sec. 3.3.2). Thus, the filter distribution $p(\mathbf{x}_t \mid \mathbf{s}_{1:t}^{(p)}, \mathbf{y}_{1:t-1})$ suffices, however the (conditionally) Gaussian joint distribution $p(\mathbf{x}_{1:t} \mid \mathbf{s}_{1:t}^{(p)}, \mathbf{y}_{1:t-1})$ could be computed straightforwardly using a Kalman smoother (e.g. running the RTS smoother backwards). Here we denote the location and covariance parameters of the Gaussian distributions corresponding to the prediction step as $m_{t|t-1}, V_{t|t-1}$, the predictive distribution (wrt. targets $\mathbf{y}_t$) as $m_{t|t}, V_{t|t}$, and the update (measurement) step as $m_t, V_t$.

The *prediction step* is given as

$$p(\mathbf{x}_t \mid \mathbf{s}_{1:t}^{(p)}, \mathbf{y}_{1:t-1}) = \int p(\mathbf{x}_t \mid \mathbf{x}_{t-1}, \mathbf{s}_t^{(p)})\, p(\mathbf{x}_{t-1} \mid \mathbf{s}_{1:t-1}^{(p)}, \mathbf{y}_{1:t-1})\, d\mathbf{x}_{t-1}$$

$$= \mathcal{N}\big(\mathbf{x}_t; m_{t|t-1}(\mathbf{s}_{1:t}^{(p)}), V_{t|t-1}(\mathbf{s}_{1:t}^{(p)})\big), \tag{17}$$

where

$$m_{t|t-1}(\mathbf{s}_{1:t}^{(p)}) = A(\mathbf{s}_t^{(p)})\, m_{t-1}(\mathbf{s}_{1:t-1}^{(p)}) + B(\mathbf{s}_t^{(p)})\, \mathbf{u}_t,$$

$$V_{t|t-1}(\mathbf{s}_{1:t}^{(p)}) = A(\mathbf{s}_t^{(p)}) V_{t-1}(\mathbf{s}_{1:t-1}^{(p)}) A(\mathbf{s}_t^{(p)})^T + R(\mathbf{s}_t^{(p)}).$$

Similarly, the *predictive distribution* (used in the *update step* below) is given as

$$p(\mathbf{y}_t \mid \mathbf{s}_{1:t}^{(p)}, \mathbf{y}_{1:t-1}) = \int p(\mathbf{y}_t \mid \mathbf{x}_t, \mathbf{s}_t^{(p)}) p(\mathbf{x}_t \mid \mathbf{s}_{1:t}^{(p)}, \mathbf{y}_{1:t-1}) d\mathbf{x}_t$$

$$= \mathcal{N}\big(\mathbf{y}_t; m_{t|t}(\mathbf{s}_{1:t}^{(p)}), V_{t|t}(\mathbf{s}_{1:t}^{(p)})\big), \tag{18}$$

where

$$m_{t|t}(\mathbf{s}_{1:t}^{(p)}) = C(\mathbf{s}_t^{(p)})\, m_{t|t-1}(\mathbf{s}_{1:t}^{(p)}) + D(\mathbf{s}_t^{(p)})\, \mathbf{u}_t,$$

$$V_{t|t}(\mathbf{s}_{1:t}^{(p)}) = C(\mathbf{s}_t^{(p)}) V_{t|t-1}(\mathbf{s}_{1:t}^{(p)}) C(\mathbf{s}_t^{(p)})^T + Q(\mathbf{s}_t^{(p)}).$$

And the *update step* yields

$$p(\mathbf{x}_t \mid \mathbf{s}_{1:t}^{(p)}, \mathbf{y}_{1:t}) = \frac{1}{Z}\, p(\mathbf{y}_t \mid \mathbf{x}_t, \mathbf{s}_t)\, p(\mathbf{x}_t \mid \mathbf{s}_{1:t}^{(p)}, \mathbf{y}_{1:t-1})$$

$$= \mathcal{N}\big(\mathbf{x}_t; m_t(\mathbf{s}_{1:t}^{(p)}), V_t(\mathbf{s}_{1:t}^{(p)})\big), \tag{19}$$

where $Z$ is the normalisation constant and where

$$m_t(\mathbf{s}_{1:t}^{(p)}) = m_{t|t-1}(\mathbf{s}_{1:t}^{(p)}) + V_{t|t-1}^T(\mathbf{s}_{1:t}^{(p)}) C^T(\mathbf{s}_t^{(p)}) V_{t|t}^{-1}(\mathbf{s}_{1:t}^{(p)})\big(\mathbf{y}_t - m_{t|t}(\mathbf{s}_{1:t}^{(p)})\big),$$

$$V_t(\mathbf{s}_{1:t}^{(p)}) = V_{t|t-1}(\mathbf{s}_{1:t}^{(p)}) - V_{t|t-1}^T(\mathbf{s}_{1:t}^{(p)}) C^T(\mathbf{s}_t^{(p)}) V_{t|t}^{-1}(\mathbf{s}_{1:t}^{(p)}) C(\mathbf{s}_t^{(p)}) V_{t|t-1}(\mathbf{s}_{1:t}^{(p)}).$$

## 7.2 SMC marginal likelihood estimate

SMC provides unbiased estimates of the marginal likelihood $p(\mathbf{y}_{1:T}) = \prod_{t=1}^T p(\mathbf{y}_t \mid \mathbf{y}_{1:t-1})$ as a by-product [9] that can be used for learning (cf. Sec. 3.3.2). The conditionals $p(\mathbf{y}_t \mid \mathbf{y}_{1:t-1})$ can be estimated by

$$p(\mathbf{y}_t \mid \mathbf{y}_{1:t-1}) = \int \int p(\mathbf{s}_{1:t-1} \mid \mathbf{y}_{1:t-1}) p(\mathbf{s}_t, \mathbf{y}_t \mid \mathbf{y}_{1:t-1}, \mathbf{s}_{1:t-1})\, d\mathbf{s}_t d\mathbf{s}_{1:t-1}$$

$$= \int p(\mathbf{s}_{1:t-1} \mid \mathbf{y}_{1:t-1}) \int p(\mathbf{y}_t \mid \mathbf{y}_{1:t-1}, \mathbf{s}_{1:t}) p(\mathbf{s}_t \mid \mathbf{s}_{t-1})\, d\mathbf{s}_t d\mathbf{s}_{1:t-1}$$

$$= \int p(\mathbf{s}_{1:t-1} \mid \mathbf{y}_{1:t-1})\, \mathbb{E}_{\pi(\mathbf{s}_t \mid \mathbf{s}_{1:t-1})}\left[\frac{p(\mathbf{y}_t \mid \mathbf{y}_{1:t-1}, \mathbf{s}_{1:t}) p(\mathbf{s}_t \mid \mathbf{s}_{1:t-1})}{\pi(\mathbf{s}_t \mid \mathbf{s}_{1:t-1})}\right] d\mathbf{s}_{1:t-1}$$

$$\approx \sum_{p=1}^P w_{t-1}^{(p)} \frac{1}{N} \sum_{n=1}^N \gamma_t(\mathbf{s}_t^{(n,p)}, \mathbf{s}_{1:t-1}^{(p)})$$

$$\approx \sum_{p=1}^P w_{t-1}^{(p)} \gamma_t(\mathbf{s}_t^{(p)}, \mathbf{s}_{1:t-1}^{(p)}) = \sum_{p=1}^P \tilde{w}_t^{(p)}.$$

The approximation of the inner importance-sampling expectation implies using a single sample (conditioned on $\mathbf{s}_{1:t-1}^{(p)}$) as is standard in SMC. The approximation wrt. the outer integral follows from $p(\mathbf{s}_{1:t-1} \mid \mathbf{y}_{1:t-1}) \approx \sum_{p=1}^{P} w_{t-1}^{(p)} \delta(\mathbf{s}_{1:t-1}^{(p)})$, using the normalised importance-weights of the previous step $w_{t-1}^{(p)} = w_{t-1}^{(p)}/\sum_{p'=1}^{P} w_{t-1}^{(p')}$.

Note that the above estimate of the conditional $p(\mathbf{y}_t \mid \mathbf{y}_{1:t-1}) = p(\mathbf{y}_{1:t})/p(\mathbf{y}_{1:t-1})$ is a ratio estimate, since the previous importance-weights are normalized by the sum of importance-weights. Although these ratio estimates are in general not unbiased, the product $p(\mathbf{y}_{1:T}) \approx \prod_{t=1}^{T} \sum_{p=1}^{P} \tilde{w}_t^{(p)}$ yields an unbiased estimator of the marginal likelihood [9, 23].

### 7.3 Innovation State Space Model

Here we describe the structure of ISSM models used for `RSGLS-ISSM`. We start by simpler exemplar instantiations of the ISSM that use a *level*, *level-trend*, or *seasonality* component. In the ISSMs considered here, the emission and control matrices $A, C$ are determined entirely by the assumed time series patterns. Only the diagonal state and observation noise covariances $Q, R$ and optionally the control matrices $B, D$ are learnable; these are computed as a weighted average of base matrices in our model. We will omit the optional matrices $B, D$ in the following for simplification.

An ISSM with only a *level* component has just a single latent variable; $A = [1]$, $C = [1]$, and noise covariances $Q = [q_1], R = [r_1]$ are positive scalars. The latent state (level) evolves over time only through innovation with additive noise, and the innovation strength is given by the (square root of the) scalar covariance $R$.

An ISSM with *level-trend* components has a 2-dimensional latent state, one representing the level and the other representing the slope of a linear trend; the model is defined by

$$A = \begin{bmatrix} 1 & 1 \\ 0 & 1 \end{bmatrix}, \quad C = \begin{bmatrix} 1 & 1 \end{bmatrix}, \quad R = \begin{bmatrix} r_1 & 0 \\ 0 & r_2 \end{bmatrix}, \quad Q = [q_1].$$

Both level and trend component evolve over time through additive noise (with covariance $R$), and the level is additionally updated with the (previous) slope of the trend. The sum of the current level and trend components are emitted (with additive noise given by scalar covariance $Q$).

ISSMs with *seasonal component* can be instantiated in several ways. Here we use the same seasonality models as in [30]. These models are described by a set of seasonal factors that assume a certain periodicity. For example, day-of-week patterns use 7 factors, one for each day of the week; similarly, hour-of-day patterns use 24 factors. Each factor can be represented by one component of the latent state. For day-of-week seasonality, we thus have a 7-dimensional latent state. The transition matrix $A$ is then the identity matrix, and the emission matrix $C = 1_{\{\mathrm{day}(t)=j\}_{j=1}^{7}}$ is an indicator (vector) that selects the component corresponding to the current day, zeroing out all other components. The noise covariance matrix $Q$ is a (positive) scalar, and $R = \mathrm{diag}\left([r_1, \ldots, r_7] \odot 1_{\{\mathrm{day}(t)=j\}_{j=1}^{7}}\right)$ is a diagonal matrix, where all components except the one corresponding to the current day are zeroed out. This is done here through element-wise multiplication (of the diagonal) with the same indicator as used for the emission matrix.

As in [30], ISSMs with multiple *seasonal components* (corresponding to different periodicities) as well as *level* or *level-trend* can be combined. The resulting transition matrix $A$ and noise covariance matrix $R$ are block-diagonal, where each block corresponds to one component. Similarly, $C$ is a concatenation of the corresponding components; consequently, the sum of the level, trend and each currently "active" seasonal component is emitted with additive noise.

In the experiments of this paper, we used a combination of *level component* and 1 or 2 *seasonal components* for the model variant `RSGLS-ISSM`: For data with daily measurement frequency (`wiki`, `exchange`), we used only day-of-week seasonality, resulting in a latent state with $7 + 1$ dimensions. In case of hourly data (`electricity`, `traffic`, `solar`), we used both hour-of-day and day-of-week seasonality, resulting in a latent state with $24 + 7 + 1$ dimensions.

### 7.4 Algorithm

The algorithm for Rao-Blackwellised particle filtering and loss computation for the ARSGLS proposed in Sec. 3 is presented in Alg. 1.

**Algorithm 1** Rao-Blackwellised particle filter with parameter estimation.

---

**Require:** $\mathbf{s}_{t-1}, \mathbf{z}_{t-1}, m_{t-1}, V_{t-1}$ are tensors with a particle, data, feature dimension. $\log \tilde{w}_{t-1}$ has particle, data dimensions. Data $\mathbf{u}_t, \mathbf{y}_t$ have data, feature dimensions. Tensors with counting indices, i.e. $\mathbf{s}_{1:T}$, have an additional time dimension that is indexed as e.g. $\mathbf{s}_t$.

1: **function** FILTER_LOSS($\mathbf{s}_{1:T}, \mathbf{z}_{1:T}, m_{1:T}, V_{1:T}, \mathbf{u}_{1:T}, \mathbf{y}_{1:T}$)
2:     $\log \tilde{w}_1, \mathbf{s}_1, \mathbf{z}_1, m_1, V_1 \leftarrow$ FILTER_STEP($\mathbf{u}_1, \mathbf{y}_1$)
3:     **for** $t \leftarrow 2 \ldots T$ **do**
4:         $\log \tilde{w}_t, \mathbf{s}_t, \mathbf{z}_t, m_t, V_t \leftarrow$ FILTER_STEP($\log \tilde{w}_{t-1}, \mathbf{s}_{t-1}, \mathbf{z}_{t-1}, m_{t-1}, V_{t-1}, \mathbf{u}_t, \mathbf{y}_t$)
5:     **end for**
6:     $\mathcal{L} \leftarrow$ compute_marginal_estimate($\tilde{w}_{1:T}$)                                    ▷ Eq. (13)
7:     **return** $\mathcal{L}$
8: **end function**

9: **function** FILTER_STEP($\log \tilde{w}_{t-1}, \mathbf{s}_{t-1}, \mathbf{z}_{t-1}, m_{t-1}, V_{t-1}, \mathbf{u}_t, \mathbf{y}_t$)
10:     initial_step $\leftarrow$ is_initial($\log \tilde{w}_{t-1}, \mathbf{s}_{t-1}, \mathbf{z}_{t-1}, m_{t-1}, V_{t-1}$)          ▷ if not provided (None)
11:     **if** initial_step **then**
12:         $\log w_t \leftarrow \log(1/P)$                     ▷ uniform weights, $P$ is the number of particles
13:         $m_{t-1}, V_{t-1} \leftarrow m_0, V_0$                              ▷ initial state prior $p(\mathbf{x}_0)$
14:         $p(\mathbf{s}_t) \leftarrow$ switch_prior($\mathbf{u}_t$)                       ▷ initial switch prior $p(\mathbf{s}_1)$
15:     **else**
16:         $\log w_t \leftarrow$ normalise($\log \tilde{w}_t$)
17:         $\log w_t, \mathbf{s}_{t-1}, \mathbf{z}_{t-1}, m_{t-1}, V_{t-1} \leftarrow$ resample($\log w_t, \mathbf{s}_{t-1}, \mathbf{z}_{t-1}, m_{t-1}, V_{t-1}$)
18:         $p(\mathbf{s}_t) \leftarrow$ SWITCH_RECURRENT_TRANSITION($\mathbf{s}_{t-1}, m_{t-1}, V_{t-1}$)          ▷ cf. below
19:     **end if**
20:     $q(\mathbf{s}_t), q(\mathbf{z}_t) \leftarrow$ encoder($\mathbf{y}_t, \mathbf{u}_t$)                              ▷ Sec. 3.3.3
21:     $\pi(\mathbf{s}_t) \leftarrow p(\mathbf{s}_t) \times q(\mathbf{s}_t)$                                    ▷ Eq. (14)
22:     $\mathbf{s}_t \sim \pi(\mathbf{s}_t)$                                    ▷ sample switch particles
23:     $\psi_t \leftarrow$ make_base_params($\mathbf{s}_t, \mathbf{u}_t$)                     ▷ base matrices $A, B, C, D, Q, R$
24:     $m_{t|t-1}, V_{t|t-1} \leftarrow$ prediction_step($m_{t-1}, V_{t-1}, \psi_t$)          ▷ Eq. (17) in App. 7.1
25:     $m_{t|t}, V_{t|t} \leftarrow$ auxiliary_predictive($m_{t|t-1}, V_{t|t-1}, \psi_t$)          ▷ Eq. (18) in App. 7.1
26:     $p(\mathbf{z}_t) \leftarrow \mathcal{N}(\mathbf{z}_t; \, m_{t|t}, V_{t|t})$
27:     $\pi(\mathbf{z}_t) \leftarrow p(\mathbf{z}_t) \times q(\mathbf{z}_t)$                                    ▷ Eq. (14)
28:     $\mathbf{z}_t \sim \pi(\mathbf{z}_t)$                              ▷ sample auxiliary particles
29:     $m_t, V_t \leftarrow$ update_step($\mathbf{y}_t, m_{t|t}, V_{t|t}, \psi_t$)                     ▷ Eq. (19) in App. 7.1
30:     $p(\mathbf{y}_t) \leftarrow$ decoder($\mathbf{z}_t$)                              ▷ Sec. 3.2
31:     $\log \gamma_t \leftarrow \log p(\mathbf{y}_t) + \log p(\mathbf{z}_t) + \log p(\mathbf{s}_t) - \log q(\mathbf{z}_t) - \log q(\mathbf{s}_t)$          ▷ Eq. (12)
32:     $\log \tilde{w}_t \leftarrow \log w_t + \log \gamma_t$
33:     **return** $\log \tilde{w}_t, \mathbf{s}_t, \mathbf{z}_t, m_t, V_t$
34: **end function**

35: **function** SWITCH_RECURRENT_TRANSITION($\mathbf{s}_{t-1}, m_{t-1}, V_{t-1}$)
36:     $F_t, S_t \leftarrow$ make_recurrent_base_params($\mathbf{s}_{t-1}$)          ▷ cond. linear state-to-switch transition
37:     $m_s, V_s \leftarrow$ marginalise_state($m_{t-1}, V_{t-1}, F_t, S_t$)          ▷ state-to-switch prediction step
38:     $p_{\mathbf{s}|\mathbf{x}} \leftarrow \mathcal{N}(\mathbf{s}_t; \, m_s, V_s)$                     ▷ Gaussian state-to-switch transition
39:     $p_{\mathbf{s}|\mathbf{s}} \leftarrow$ switch_transition($\mathbf{s}_{t-1}, \mathbf{u}_t$)               ▷ Gaussian switch-to-switch transition
40:     $p(\mathbf{s}_t) \leftarrow$ gaussian_linear_combination($p_{\mathbf{s}|\mathbf{x}}, \, p_{\mathbf{s}|\mathbf{s}}$)          ▷ sum of means and covariances
41:     **return** $p(\mathbf{s}_t)$
42: **end function**

---

## 7.5 Further results

Here we provide further results for `ARSGLS` on `wiki` and `Box` datasets, evaluating the impact of the encoder and the number of particles during training. Furthermore, we provide pixel accuracy results for the experiments described in Sec. 5.3. On `Box`, we run each experiment with 3 different seeds and for `wiki`, we run with 4 random seeds as in the main text. Note that for `wiki` we use a different initialisation scheme compared to the experiments in the main text (Xavier instead of the default in Pytorch 1.6), resulting in better scores.

### 7.5.1 Encoder and proposal distribution

Figure 5: Pixel accuracy (left) and Wasserstein distance (right) on `Box` with/without encoder.

Here we show the importance of the encoder that approximates the likelihood function by a Gaussian (cf. Sec. 3.3.3). We conducted experiments where we omit the encoder and thus use a bootstrap proposal distribution. On `wiki`, using 32 particles during training without encoder yields CRPS scores $0.265 \pm 0.010$ for rolling evaluation and $0.360 \pm 0.006$ for non-rolling (long-term) evaluation, respectively. In contrast, the same number of particles with encoder yields significantly better CRPS scores of $0.207 \pm 0.000$ and $0.263 \pm 0.003$, respectively.

Similarly, the accuracy and Wasserstein distance for the `Box` dataset is significantly better when using our ladder-type encoder, as shown in Fig. 5. Without encoder for the proposal distribution, the model is not able to learn the dynamics and fails to converge to a reasonable fit.

## 7.6 Number of particles

We evaluate our model for a different numbers of particles $(1, 8, 16, 32, 64, 96)$ used during training. The performance on the `wiki` dataset is shown in Fig. 6. Surprisingly, the forecasting performance does not improve with more particles.

Figure 6: CRPS scores on `wiki` dataset for varying number of particles used for training.

In contrast, Fig. 7 shows that using more particles leads to a significantly better forecasting performance on the `Box` dataset as expected.

Figure 7: Accuracy (left), Wasserstein distance (right) on `Box` for varying number of particles used for training.

Figure 8: Pixel accuracies in filter/smoothing (`ARSGLS`/ `KVAE`) range ($t < 20$) and forecast range ($t \geq 20$). Results are averaged over 1000 test samples and 3 independent runs.

### 7.6.1 Simulated physical environments

Additional to the the Wasserstein distance reported in the main text, we provide pixel accuracies in Fig. 8

## 7.7 Experiment details

### 7.7.1 Pendulum

The model used in the pendulum experiment has $3$ and $5$ state and switch dimensions, respectively. We used $10$ base matrices (for each of $A, C, Q, R$ and additionally $F, S$ in case of the recurrent model); the weights are predicted by an MLP with 1 hidden layer of 32 units with leaky relu activations (and a softmax output). All covariance matrices ($R, Q, S$) are diagonal and represented as log of the inverse scale to ensure positive variance. State and switch prior $p(\mathbf{x}_0)$, $p(\mathbf{s}_1)$ are both trainable diagonal Gaussians (scale parameters are again represented as log inverse scale), initialised as the standard Normal. The switch transition function $f(\mathbf{s}_{t-1})$ (cf. Eq.(9)) is an MLP with 1 hidden layer of 32 units and leaky relu activations; no encoder (cf. Sec.3.3.3) is used in this experiment.

Both models are trained using the Adam optimizer with $\beta_1 = 0.9$, $\beta_2 = 0.95$, and (initial) learning rate $1 \times 10^{-2}$ with an exponential decay to $1 \times 10^{-4}$ over a total of $50$ epochs. The batch size is $100$, $P = 64$ particles are used for learning and $P = 100$ particles for computing the empirical mean and std. deviation of the GMM (cf. Eq. (8)) filter and forecast distribution in the evaluation plot.

### 7.7.2 Univariate time series forecasting

A short summary of the datasets, as used in [32], considered for these experiments are given:

- `electricity`: hourly electricity consumption of 370 customers;
- `traffic`: hourly occupancy rates of 963 car lanes of San Francisco bay area freeways;
- `solar`: hourly photo-voltaic production of 137 stations;
- `exchange`: daily exchange rate of 8 currencies;
- `wiki`: daily page view of 2000 Wikipedia pages.

For all datasets, the same model and training hyper-parameters (except learning rate) were used. Furthermore, `RSGLS-ISSM` and `ARSGLS` use the same model architecture where possible. For these datasets, each model are given time-features (hour of day, day of week, etc.) and time-series indicators as inputs $\mathbf{u}_{1:T}$ as in [30, 33]. Time-series indicators are embedded in $50$ dimensions (except 8 dimensions for `exchange` dataset), and combined with the time-features by a single neural network layer with $64$ units and leaky relu activations.

The state dimension of `RSGLS-ISSM` is determined by the ISSM structure (cf. App.7.3), whereas for `ARSGLS` the state has 16 dimensions. The switch $\mathbf{s}$ and auxiliary variable $\mathbf{z}$ (in case of `ARSGLS`) have 10 dimensions each. Furthermore, 20 base matrices are used. In case of `RSGLS` these include only

$D, Q, R, F, S$ since $A, C$ are determined by the ISSM. `ARSGLS` uses additionally learnable matrices $A, C$. For both models $B$ was not used. The weights for averaging the matrices are predicted by an MLP with 1 hidden layer of $64$ units with leaky relu activations and a softmax output (taking $\mathbf{s}_t$ as input). Covariance matrices $(R, Q, S)$ are diagonal and parametrised as the log of the inverse scale.

The state prior $p(\mathbf{x}_0)$ is a trainable diagonal Gaussian, initialised as the standard Normal; the switch prior $p(\mathbf{s}_1 \mid \mathbf{u}_1)$ is a diagonal Gaussian for which the location and scale parameters are predicted by a linear transformation (and additional softplus in case of scale), taking the $64$ input features from the embedding and subsequent neural network layer (see above) as inputs. Similarly, the switch transition function $f(\mathbf{s}_{t-1}, \mathbf{u}_t)$ (cf. Sec. 3.1) is an MLP with 1 hidden layer of 64 units and leaky relu activations, which takes both the $64$ input features and the previous switch as inputs.

Encoders differ for `RSGLS-ISSM` and `ARSGLS`, respectively, since the latter has the additional auxiliary variable (cf. Sec. 3.2). However, in both cases the proposal distribution is formed as a product of Gaussians as described in Sec. 3.3.3. For `RSGLS-ISSM`, the encoder is a diagonal Gaussian for which the parameters are predicted by an MLP with 1 hidden layer of 64 units and leaky relu activations (outputs corresponding to the scale use softplus activations). The additional non-linear emission model of `ARSGLS` is a diagonal Gaussian for which the parameters are predicted by an MLP with 2 hidden layers with 64 units and leaky relu activations. The ladder encoder shares an MLP that predicts the parameters of the Gaussians corresponding to the auxiliary and switch variable, respectively. This shared MLP has 3 hidden layers with leaky relu activations; the parameters of the Gaussians are predicted from the 2nd and 3rd (last) hidden layer, respectively.

Each model is trained using the Adam optimizer with $\beta_1 = 0.9$, $\beta_2 = 0.95$. For `solar`, the initial learning rate is $1 \times 10^{-2}$, for `electricity` $1 \times 10^{-3}$ and for all other datasets $5 \times 10^{-3}$. In each experiment, the learning rate is decayed over a total of 2500 iterations by a factor $10^{-2}$. The batch size is $50$ and $P = 10$ particles are used.

### 7.7.3 Simulated physical environments

Model architectures (for components that are similar between both models) are chosen as in [11], except that the state dimension and number of base matrices is $10$ for both, instead of 4 and 3, respectively. Furthermore, the (diagonal) covariance matrices $R, Q$ (parameterised as log inverse scale, i.e. $\log R^{-1/2}$) are learnable, whereas in [11] these are fixed hyperparameters.

The switch in `ARSGLS` has $8$ dimensions, and the switch prior is a trainable diagonal Gaussian, initialised as the standard Normal. The auxiliary variable has 6 dimensions for Pong (to encode the position of the ball and both pads) and 2 dimensions for all other datasets as in [11]. The switch transition function $f(\mathbf{s}_{t-1})$ from Eq. (9) is an MLP with 1 hidden layer of $64$ units and relu activations. The ladder encoder of `ARSGLS` uses the same convolutional architecture for the auxiliary variable as the encoder in `KVAE`, and an additional hidden layer with 64 units for the switch. In contrast to the `KVAE`, the Gaussian from the encoder is combined with the respective Gaussian of the generative model as a product of these densities (cf. Sec. 3.3.3).

The same optimisation hyper-parameters are used for training as in [11], with the exception that we train each model for $400$ epochs with an initial learning rate of $0.002$, decaying every 20 epochs by $0.85$ (instead of $80$ epochs with initial learning rate $0.007$ with the same decay). The number of particles in `ARSGLS` is $P = 32$.