[Reviews · NeurIPS 2020]

Review 1

Summary and Contributions: This paper proposes an extension to state-switching Gaussian linear systems, which extends the ideas of traditional Kalman filtering to the setting of switching states. One of the main differences from this work compared to others is that it uses Gaussian switch variables which allows the authors to leverage the conditionally linear-Gaussian aspect of the model to make inference more efficient through the use of Rao-blackwellised particle filters. This model is easily extended to the setting of non-Gaussian, nonlinear observations and can handle multivariate data. The authors assess the effectiveness of their model on three examples and compare to popular competing methods that have previously been published at NeurIPS and similar ML conferences.

Strengths: - The method is theoretically sound and mathematically correct. - The simulation study considers three different models, one of which tests the method on five real-world datasets. The authors algorithm is compared against suitable, alternative models and shows that it is competitive. - The paper is not ground-breaking in its contribution, but does offer an important contribution to the work in deep state-space models. Particularly, because this work takes advantage of the conditionally linear-Gaussian aspect of the model which allows the authors to use efficient proposal distributions in their SMC through Rao-Blackwellisation, which is well known in the SMC community to significantly reduce the variance in the particle weights. It would have been interesting for the others to have considered simpler proposals, e.g. bootstrap, so to test how much of the improvement in their results follows from the Rao-Blackwellisation.

Weaknesses: - Potentially, one of the drawbacks of this approach is that it is restricted to linear latent transitions, which the authors do not really comment on. The authors have made clearer that their approach could be used for nonlinear models by linearising the dynamics and utilising the approximate linearised form. - It isn't clear what level of improvement is offered by models such as DeepAR, KVAE and the authors' model over simpler, non-deep models. It would have been helpful for the authors to have included these (as a baseline) in Sections 5.2 and 5.3.

Correctness: - The methodology is mathematically correct and the additional derivations in the supplementary material help to clarify some of the details in the main paper. - The approach taken to test the algorithm in the experiments section is sound. - The authors highlight that their proposal distribution is of a "similar structure" to the optimal proposal distribution. It isn't clear how close to optimal this is, or whether under certain model settings the optimal proposal exists? - The authors fit their model using SGD on a lower bound to the log-marginal likelihood. Why not directly optimse the log-likelihood itself as was done in Poyiadjis et al. (2011)? It's not clear how tight this lower bound is, and therefore how accurate the parameter estimation will be.

Clarity: -The paper is very clearly written. The introduction highlights the important contributions of this work and how the paper fits with other existing deep SSM models. -The authors stick with standard SSM notation, which makes it a lot easier for readers who are already familiar with this notation. - It would have been helpful for the reader if the authors had included an algorithm box (perhaps in the supplementary material if space is an issue) that would clearly outline the steps of their algorithm. - The densities p(x_t|x_{t-1}) and p(y_t|x_t) are not defined when introduced on page 2. Additionally, the Dirac function in (1) is also not defined. - From eq. (5) onwards there are various points where the authors use a diagonal line in the conditional density to show that these terms can be removed, however, this is never explained as to why these terms can be dropped. Is it because they are independent, or are the authors making an assumption so that dropping these terms makes everything easier?

Relation to Prior Work: Section 4 of the paper clearly outlines how this work differs from existing work

Reproducibility: Yes

Additional Feedback: In Section 3.3.3 - it is stated that q(z_t,s_t|y_t) \apporx p(y_t|z_{1:t},s_{1:t}) it isn't clear why this is true.


Review 2

Summary and Contributions: This paper proposes an improved switching linear Gaussian state space model, which uses continuous switching variables and an auxiliary variable in the non-linear emission model. The state dynamics are linear, making them tractable to evaluate, while other components of the model are still expressive. The authors use particle filtering and a sequential Monte Carlo objective. The experiments compare the model against a few relevant baselines on a variety of sequence datasets, as well as simple videos of objects. The model performs competitively in all cases. Update: After reading the authors' response and the other reviews, I am deciding to maintain my score. I feel that this paper presents a reasonable contribution, which is presented and demonstrated well. Accordingly, I suggest acceptance.

Strengths: Soundness: The model formulation appears to be mathematically sound. As with several previous works, the authors utilize linear-Gaussian distributions for dynamics, which have the benefit of permitting exact computation of expectations, e.g. Kalman filtering. The authors propose two main improvements over related models: 1) the use of recurrent switch transitions through Gaussian switch variables, and 2) non-linear emission models through the use of an additional (auxiliary) latent variable, z. They train this model with a sequential Monte Carlo objective utilized in previous works. This paper builds off of many of the theoretical developments of previous works, adding a couple of useful techniques. Throughout, the mathematical formulation appears to be rigorous and well explained. The empirical evaluation appears to be broadly sound as well. The authors evaluate their model against two related deep state space models with linear dynamics, as well as a competitive RNN baseline. The model is demonstrated on a toy example, then evaluated on a set of sequence datasets as well as a set of simple videos from simulated physical environments. The model is evaluated in each setting using metrics that quantify the predictive performance. In all cases, the model is competitive. While this analysis could possibly be extended to other data domains, the present evaluation appears sufficient to conclude that the model is capable of accurately estimating dynamics. Significance: Dynamics estimation is an important aspect of many scientific and engineering disciplines. This paper helps to show that there are benefits to using dynamics models with simple (i.e. linear Gaussian) forms to improve estimation. While this paper is not the first to make this point, the authors compare with the two most relevant previous works in the area, demonstrating performance improvements in multiple domains. While it’s not clear that the techniques in this paper fundamentally change dynamics modeling, the paper is nevertheless a useful contribution and could provide a starting point for future works. Novelty: There are a lot of moving parts in this model, and while I have read many of the previous works in this area, I found it difficult to assess the exact novelty of each contribution. From what I understand, the novelty of this work comes from the use of Gaussian switch variables to help parameterize dynamics in conjunction with the particle filtering objective and auxiliary latent variable. Each of these contributions has been developed separately, but this work combines these ideas to form a single model. While the methodologies may not be particularly novel, combining these ideas is non-trivial, and the authors demonstrate the empirical benefits of this approach. Relevance: Dynamics estimation, particularly with deep probabilistic models, is a core area of interest in the NeurIPS community. The contributions of this paper seem fairly well aligned with many previous works that have appeared at NeurIPS.

Weaknesses: Soundness: I did not spot any particular weaknesses in terms of the theoretical formulation, however, there may be aspects of the model and training procedure that I’m missing. One question I have is whether the benefits of the linear dynamics can be extended to hierarchies of state variables. While I appreciate the mathematical rigor in this paper’s formulation, it is unclear to me whether such techniques can be scaled to larger models. I’m not familiar with the datasets used in the evaluation, though they have been used in some previous works. My concern here is whether these datasets are too simple, though previous works by Krishnan et al. and Fraccaro et al. also evaluated their models on similar sequential settings. It may help to expand the experiments to include other benchmark datasets. The authors also use two evaluation metrics with which I am unfamiliar. While I understand that these quantify predictive performance, it may be helpful to also report more standard metrics, like log-likelihood. Significance: Although this paper demonstrates performance improvements over previous models, it’s not clear that these benefits outweigh the complexity of the model. For instance, an RNN baseline outperforms all models on multiple datasets. Although the model does seem to perform well at long-term prediction, additional long-term prediction results (e.g. generated videos) may help to demonstrate the benefits of this dynamics parameterization. Novelty: As stated above, this paper effectively combines multiple previously developed techniques into a single model and training setup. While I do not necessarily view this as a weakness of the approach, it is less novel because of this. Relevance: Much of the NeurIPS community is focused on vision, audio, text, etc. domains. While this paper includes experiments on video, the sequences are fairly simple. In this regard, the paper will likely only appeal to a subset of the broader NeurIPS community.

Correctness: Yes, the claims and method appear to be correct. The authors provide a fairly rigorous formulation of their model and inference procedure. The empirical methodology also appears to be correct and rigorous. I’m unclear about the degree to which the presented evaluation criteria are standard in the dynamics modeling community. Log-likelihood results could help to supplement the current evaluation.

Clarity: Overall, the paper is very clear and well written. My main concern is that there are many moving parts to the model and inference procedure. Yet, there are no diagrams depicting the model, both in the main paper and the supplementary. It would be helpful to include diagrams showing the structure of the model, as well as possibly visualizing the inference procedure. This would also be helpful to distinguish the proposed model from previous models. An algorithm box (in the appendix) would also help to clarify the overall picture.

Relation to Prior Work: The authors provide a discussion on the relation to previous work in Section 4. As noted above, some type of model diagram could help to illustrate the architectural differences between the various models that have combined deep networks with linear-Gaussian dynamics. The authors may also want to mention other works in the general family of deep sequential latent variable models, such as VRNN (Chung et al., 2015), SRNN (Fraccaro et al., 2016), etc. I believe the authors may want to cite Le et al., Auto-encoding sequential monte carlo, 2018 in addition to Maddison et al. and Naesseth et al. The authors discuss comparing with Deep State Space Models and Kalman VAE in the experiments section, but it would be helpful to cite these works explicitly here.

Reproducibility: Yes

Additional Feedback: My primary recommendation is to visualize the model diagram. Given the number of different components in the model, this paper needs some sort of visualization. Figure 1: Should label or explain that the blue and orange curves correspond to two different trajectories. Initially, I assumed blue and orange were from two different models. Figure 3: The labels should be enlarged.


Review 3

Summary and Contributions: This work addresses two draw-backs of Switching Gaussian Linear Systems (SGLS). 1) long-term forecasting, 2) complex multivariate observations SGLS use latent variables s_t to switch between different Gaussian linear models, which have states x_t, and observations y_t. To handle longer range dependencies (the first problem) the authors add a linear state-to-switch dependence to the basic framework of SGLS, by introducing an additional latent variable s_t with transition probability p(s_t | s_{t-1}, x_{t-1}). For the second problem (allowing for a more complex observation model), the authors add one auxiliary variable z_t to construct flexible conditional distributions p(y_t | z_t) which are modeled by a non-linear emission model (e.g. neural network). Since the adding state recurrence is linear in Eq.(8), using a particle filter for the posterior approximation is still tractable. Once adding the non-linear emission model, they need to optimize the tractable lower-bound as in variational sequential monte carlo.

Strengths: Significance: This work addresses a relevant problem. I liked the derived particle filter. It made use of the structure of the model to do as many compuations as possible in closed form.

Weaknesses: Novelty: The idea of adding auxiliary variable and the non-linear emission model to GLS is already proposed by Kalman-VAE. Adding the recurrence of state might be new, but the general idea is somehow well-known. Therefore, the novelty is limited in my opinion. The experimental results are not very strong. Especiallz, the first experiment on the pendulum is not too convincing. Although the pendulum dynamics are non-linear, previous work using locally linear transitions (e.g. deep variational bayes filters, recurrent kalman networks) also show very good results. It would have been more convincing if the comparison included at least one competitive previous work, not only SGLS.

Correctness: Yes.

Clarity: Yes.

Relation to Prior Work: Yes.

Reproducibility: Yes

Additional Feedback:

[Author Response · NeurIPS 2020]

We thank the reviewers for their invested time and valuable suggestions, which we will gladly incorporate to improve
the paper. We first respond to two comments made by all reviewers and then address additional comments separately.

**Novelty**: As pointed out by the reviewers, the proposed model and algorithm have several components that are similar
to previously published work. While we did attempt to highlight the novel contributions and the overlap with prior
work in the related work and methods sections, all reviewers raised questions around this point, so we provide further
clarification about these similar components below and we will update the relevant sections to point out the novelty.
i) **Non-linear parametrisation of a GLS**: The KVAE and DeepState parametrise the GLS through a *deterministic*
RNN and use inputs that have only partial information to predict the SSM parameters. That is, DeepState uses only
controls $\mathbf{u}_t$ (open loop) and KVAE uses *samples* of the *pseudo-observations* $\mathbf{z}_t$. In contrast, our *probabilistic* switches
transition is conditioned on the controls $\mathbf{u}_t$ and the state variable $\mathbf{x}_t$ for which uncertainty is preserved (no sampling)
and which includes dynamics information (in contrast to $\mathbf{z}_t$). Furthermore, we perform *probabilistic inference* via
(Rao-Blackwellised) SMC rather than assuming that the SSM parameters can be predicted with absolute certainty.
ii) **State-to-switch recurrence**: While previous work (Barber 2006, Linderman 2017, Becker-Ehmck 2019) has also
used a form of recurrence, the details differ substantially. The approach of Linderman 2017 is unfortunately not
suitable for optimization with SGD since the Polya-Gamma distribution can not be reparametrised. Becker-Ehmck
2019 samples the state variable $\mathbf{x}_t$ and Barber 2006 makes a sample-based or deterministic approximation. In contrast,
our conditionally linear Gaussian state-to-switch recurrence allows us to maintain the full distribution.
iii) **Auxiliary variable with a decoder-type neural network**: Our approach differs from the same component in the
KVAE in the inference algorithm and choice of proposal distribution / variational approximation. Whereas the KVAE
uses an encoder-based variational approximation, we apply SMC and use a similar encoder only as part of the proposal
distribution, i.e. by taking the product with the conditional distribution $p(\mathbf{z}_t, \mathbf{s}_t | \mathbf{z}_{1:t-1}, \mathbf{s}_{1:t-1})$ (where the states $\mathbf{x}_{t-1}$
are marginalised out). This is well motivated by the structure of the optimal proposal distribution (see comments below)
and similar arguments could have been made for the variational approximation in the KVAE.

**Algorithm box and visualisation**: These are good suggestions, we will use the extra page to add an algorithm box
with references to a visualisation of the graphical model and the computational structure of the inference procedure.

**Reviewer 1**:
i) We agree that evaluating the **impact of the Rao-Blackwellisation** is interesting and will provide additional results.
ii) Regarding the potential **drawback of the linear latent transitions**, note that *conditionally* linear systems can
approximate non-linear systems through linearisation around the current state. We will expand on this in the paper.
iii) Wrt. the **optimal proposal distribution**, cf. Sec. 3.3.3 and the numerator of Eq. (11). The latter two conditionals
are known; we therefore choose the same factorisation and only replace the likelihood term by a Gaussian approximation
(through an encoder). It is optimal if the likelihood is indeed Gaussian. We will discuss this in more detail.
iv) We were not aware of Poyiadjis et al. (2011). This gradient estimate seems related to black-box variational inference
in the sense that both are based on a log-derivative trick. It potentially yields higher-variance estimates, so there might
be a variance-bias trade-off. This is an interesting research question that we try to answer for the camera-ready version.
v) Thank you for carefully spotting undefined quantities and unclear statements, we will clarify these in the paper. All
of the crossed (diagonal line) terms cancel due to independence, no additional assumptions are made.

**Reviewer 2**:
i) It is indeed possible to extend the approach to **hierarchies of state variables**, e.g. by interleaving non-linear, sampled
switches with linear states. The details are not as straightforward, but we plan to develop this extension in future work.
ii) **Scaling the approach to larger models** should be feasible. The decoder-type likelihood in Sec. 5.3 already uses a
CNN and the non-linear switch transitions could also make use of more recent architectures that use gating or attention,
although the latter requires to alleviate the Markovian assumption.
iii) Regarding the **strong baseline DeepAR**, we will add a short motivation/discussion about other practical benefits of
SSMs compared to AR models, e.g. in applications such as anomaly detection and when handling missing data. AR
models need to impute missing or anomalous data and thus accumulate errors. In contrast, SSMs maintain uncertainty
information and provide a principled way to ignore missing or anomalous data, i.e. by omitting the Bayes update
(Kalman update step). Note also that DeepAR as originally described can handle only univariate time-series and can
thus not be directly applied to the scenario from Sec. 5.3.
iv) We will discuss the related work you pointed out which is indeed very relevant.

**Reviewer 3**:
i) Please see above for detailed comments on the **novelty**.
ii) The purpose of the **pendulum toy experiment** in Sec. 5.1 is to qualitatively demonstrate the necessity of the
state-to-switch recurrence. For a quantitative evaluation and comparison with related methods please refer to Secs. 5.2
and 5.3. We agree that a comparison with DVBF, DKN and others would have been interesting, but we decided to focus
on the most closely related methods, i.e. KVAE and DeepState.

[Meta-Review · NeurIPS 2020]

This paper proposes an extension to state-switching Gaussian linear systems, which extends the ideas of traditional Kalman filtering to the setting of switching states. Strengths: - the use of recurrent switch transitions through Gaussian switch variables is interesting - mathematical elegance is appealing Weaknesses: - It is unclear if the proposed techniques can be scaled to larger models - video experiments could be strengthened